# Room-temperature single-photon emission from β-Ga₂O₃

Yiming Shi[1,2,4], Zhengchang Xia[2,3,4], Junhua Meng [ORCID][1] ✉, Libin Zeng[2,3], Ji Jiang [ORCID][2], Zhouxin Li[2,3], Aoxing Wang[2,3], Huabo Yang[2,3], Zhigang Yin[2,3] & Xingwang Zhang [ORCID][2,3] ✉

Single photon emitters (SPEs) hosted by the wide bandgap semiconductors have the great potential to enable quantum applications at room temperature. Recently, many defect-based SPEs have been discovered in various wide bandgap materials, such as diamond, AlN, SiC, h-BN, GaN and ZnO. Beta-phase gallium oxide (β-Ga₂O₃) is an emerging ultrawide bandgap semiconductor with promising electronic and optoelectronic properties, however, there has been no report on single-photon emission from β-Ga₂O₃ to date. Herein, we present the demonstration of room-temperature photostable single-photon emission from β-Ga₂O₃. We find that the SPEs can be found in a variety of β-Ga₂O₃ including homoepitaxial and heteroepitaxial β-Ga₂O₃ films and commercially available β-Ga₂O₃ wafers. The observed emitters have excellent photophysical characteristics including high purity, high brightness, and linear polarization. First-principles calculations predict that a localized neutral divacancy defect, generated by plasma treatment and activated by annealing, is responsible for the SPEs in β-Ga₂O₃. The high-performance room-temperature SPEs embedded in a technologically mature semiconductor are promising for on-chip scalable integrated devices and quantum technologies.

Single photon sources are key elements for various scalable quantum information technologies, including quantum communication, quantum computing, quantum sensing and measurement[1–3]. So far, single photon emission has been observed from various quantum emitters such as single molecules[4], atoms[5], ions[6], quantum dots[7], and point defects in semiconductors[8,9]. Among them, color centers in wide bandgap semiconductors are considered to be one of the most promising single photon emitters (SPEs) due to their unique properties of stable emission at room temperature or even at high temperature, as well as a wide range of single-photon emission wavelengths[9]. Furthermore, the SPEs based on wide bandgap semiconductors are compatible with mature semiconductor technologies, which is beneficial for the integration of optoelectronic devices. In recent years, defect-based SPEs have been discovered in various wide bandgap

materials, such as diamond[10–12], silicon carbide (SiC)[13–15], silicon nitride (SiN)[16], zinc oxide (ZnO)[17,18], gallium nitride (GaN)[19,20], aluminum nitride (AlN)[21,22], and hexagonal boron nitride (h-BN)[23–26].

Beta-phase gallium oxide (β-Ga₂O₃) has emerged as a potentially disruptive ultra-wide bandgap semiconductor for next-generation electronic and optoelectronic applications[27–30], as it has a wide bandgap of ~4.9 eV, high breakdown electric field of 8 MV cm⁻¹, excellent chemical and thermal stability, and large Baliga's figure of merit of 3444. To date, β-Ga₂O₃ has well-established growth and device engineering protocols, and β-Ga₂O₃ wafers up to 8-inch are now commercially available. Unlike traditional semiconductors, β-Ga₂O₃ possess a low-symmetry monoclinic crystal structure (C2/m space group), characterized by two crystallographically nonequivalent Ga³⁺ and three nonequivalent O²⁻ ions in the unit cell. The abundance of

¹School of Physics and Optoelectronic Engineering, Beijing University of Technology, Beijing, PR China. ²State Key Laboratory of Semiconductor Physics and Chip Technologies, Institute of Semiconductors, Chinese Academy of Sciences, Beijing, PR China. ³Center of Materials Science and Optoelectronics Engineering, University of Chinese Academy of Sciences, Beijing, PR China. ⁴These authors contributed equally: Yiming Shi, Zhengchang Xia. ✉e-mail: jhmeng@bjut.edu.cn; xwzhang@semi.ac.cn

possible natural defects makes β-Ga$_2$O$_3$ a flexible and scalable material platform for defect-based SPEs. Recently, Stehr et al. reported a transition-metal color center in β-Ga$_2$O$_3$ that emits in the telecom range and has an electronic structure suitable for quantum information applications[31]. However, there are no reports on single-photon emission from β-Ga$_2$O$_3$ yet.

In this work, we demonstrate the single-photon emission from β-Ga$_2$O$_3$ at room temperature. The samples we used are homoepitaxial β-Ga$_2$O$_3$ films, heteroepitaxial β-Ga$_2$O$_3$ films on sapphire substrates, and bulk β-Ga$_2$O$_3$ wafers, which were irradiated with plasma then annealed at high temperatures to generate SPEs. Second-order correlation measurements reveal strong photon antibunching, which unambiguously establishes the single-photon nature of the emission. The SPEs in β-Ga$_2$O$_3$ exhibit high purity and brightness, excellent stability, as well as spatially uniform distribution. The prevalence of SPEs in different β-Ga$_2$O$_3$ samples indicates that the intrinsic defects are responsible for the observed quantum emissions. We performed first-principles calculations to identify the possible origin of the observed SPEs. This work provides a reliable and scalable platform for further technological development, which may open promising avenues for photonic quantum devices based on β-Ga$_2$O$_3$.

## Results
### Characterization of epitaxial β-Ga$_2$O$_3$ thin films

The homoepitaxial and heteroepitaxial β-Ga$_2$O$_3$ thin films were grown by low pressure chemical vapor deposition (LPCVD) on single crystal β-Ga$_2$O$_3$ (−201) and sapphire substrates, respectively. The atomic force microscopy (AFM) image (Fig. 1a) demonstrates that the surface of the polished β-Ga$_2$O$_3$ substrates is rather smooth and uniform with a root mean square (RMS) roughness of $0.48 \pm 0.04$ nm (Supplementary Fig. 1a). The homoepitaxial β-Ga$_2$O$_3$ film exhibits a clear step-flow growth mode (Fig. 1b) with an RMS roughness of $2.5 \pm 0.31$ nm

(Supplementary Fig. 1b), which is in good agreement with the previously reported β-Ga$_2$O$_3$ homoepitaxial films[32,33]. In contrast, the heteroepitaxial β-Ga$_2$O$_3$ film grown on a sapphire substrate follows the island-growth model and displays coral-like morphology (Fig. 1c) with a relatively large RMS roughness of $3.1 \pm 0.22$ nm (Supplementary Fig. 1c). Both Raman spectra of the homoepitaxial and heteroepitaxial β-Ga$_2$O$_3$ films show a set of characteristic Raman peaks at 113.2, 143.8, 168.9, 199.3, 319.7, 345.6, 416.0, 657.5 and 765.3 cm$^{-1}$ (Fig. 1d), which are well consistent with the previously reported data[32,34]. Furthermore, X-ray photoelectron spectroscopy (XPS) measurements confirm that the homoepitaxial β-Ga$_2$O$_3$ films have high chemical purity and their chemical composition is close to the ideal stoichiometry (Supplementary Fig. 2). The thickness of epitaxial β-Ga$_2$O$_3$ film is about 200 nm, as revealed by the low magnification cross-sectional transmission electron microscopy (TEM) image (Supplementary Fig. 3).

We conducted high-resolution TEM (HRTEM) measurements to investigate the interfacial microstructure and crystalline quality of the homoepitaxial β-Ga$_2$O$_3$ films, taken along the zone axis <102> of β-Ga$_2$O$_3$ (Fig. 1e). The atomically sharp interface between the epitaxial layer/substrate almost cannot be distinguished (the interface indicated by the dashed line is determined from the global large-scale TEM image in Supplementary Fig. 4), revealing the high-quality homoepitaxy of the β-Ga$_2$O$_3$ film. As shown in the magnified interfacial region (Fig. 1f), the $d$-spacing of lattice structure is 4.68 Å, matching the (−201) planes of β-Ga$_2$O$_3$, while the lateral lattice distance of β-Ga$_2$O$_3$ is about 1.52 Å, corresponding to the lattice spacing of β-Ga$_2$O$_3$ (020) planes. These results are aligning with the observed sharp and bright diffraction spots of the selected area electron diffraction (SAED) pattern taken from the interface (the inset of Fig. 1e). In addition, the interface structure of the heteroepitaxial β-Ga$_2$O$_3$ film was also investigated (Supplementary Fig. 5), and the epitaxial relationship between the β-Ga$_2$O$_3$ epilayer and the sapphire substrate is determined as

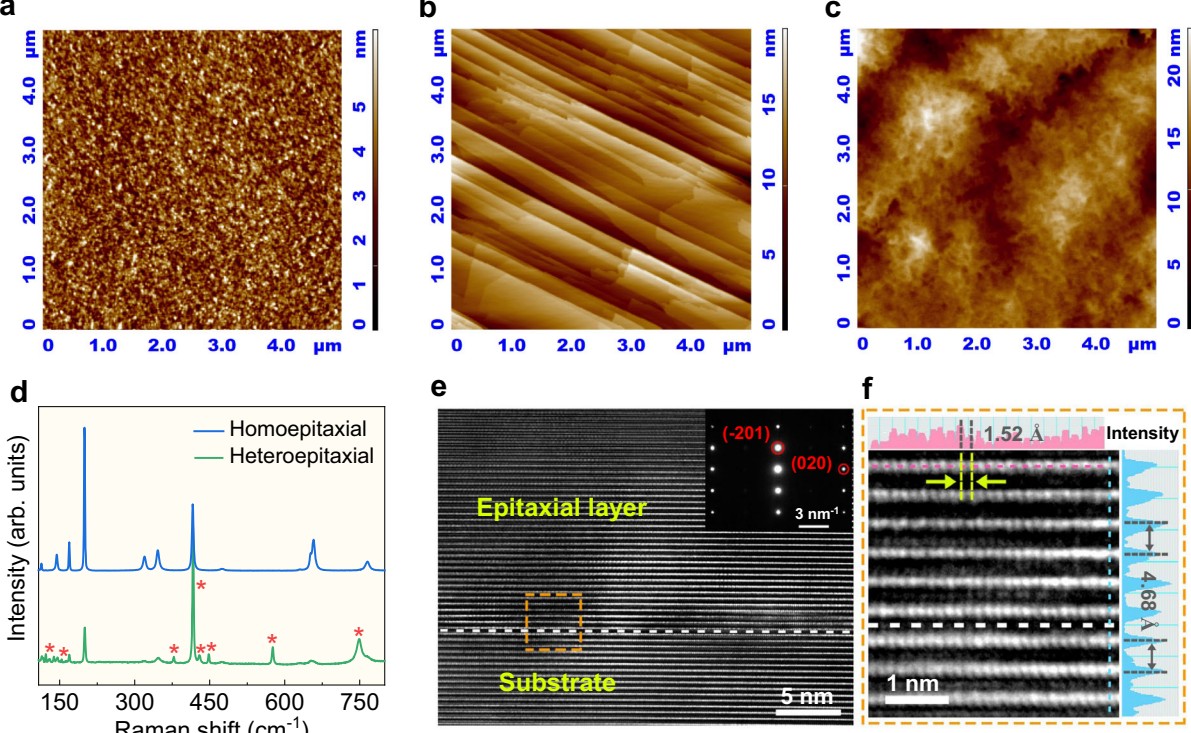

**Fig. 1 | Characterizations of the epitaxial β-Ga$_2$O$_3$ thin film.** Atomic force microscopy (AFM) images of the single crystal β-Ga$_2$O$_3$ wafer (**a**), the homoepitaxial β-Ga$_2$O$_3$ (**b**) and heteroepitaxial β-Ga$_2$O$_3$ (**c**) films grown on sapphire substrate. **d** Raman spectra of the homoepitaxial and heteroepitaxial β-Ga$_2$O$_3$ films. The asterisks denote the Raman peaks of sapphire substrate. **e** High-resolution transmission electron microscopy (HRTEM) image of the homoepitaxial β-Ga$_2$O$_3$ interface taken along the β-Ga$_2$O$_3$ [102] direction. **f** A magnified view of the orange boxed area in (**e**) and the corresponding atomic intensity profile along the vertical dash lines.

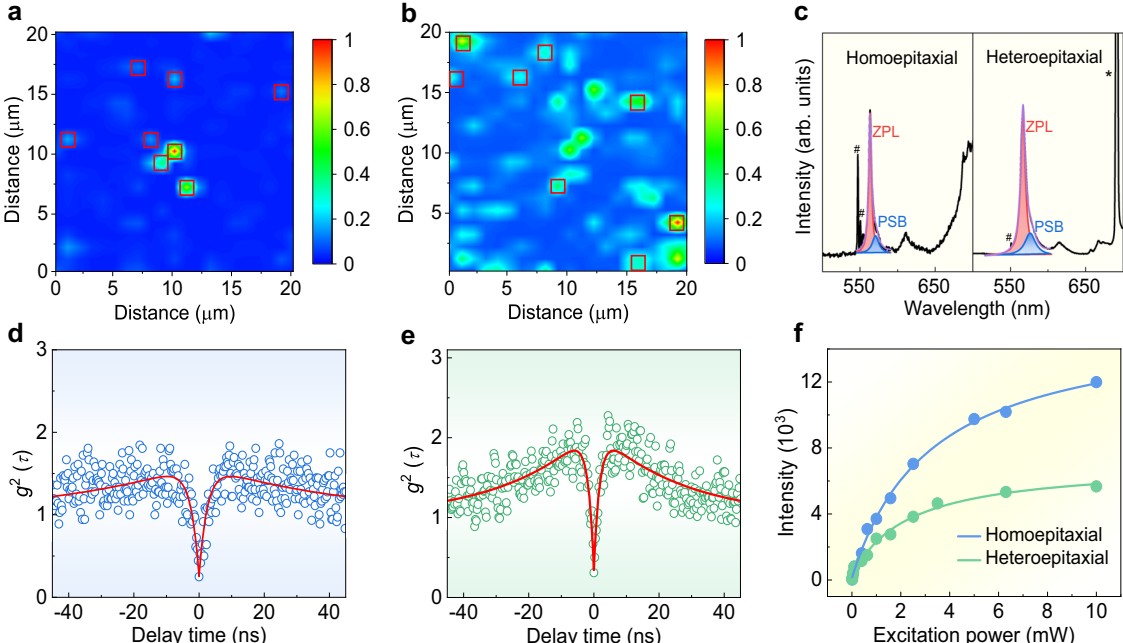

**Fig. 2 | Characterization of room-temperature SPEs in β-Ga₂O₃.** Normalized confocal photoluminescence (PL) intensity maps of the homoepitaxial β-Ga₂O₃ (**a**) and heteroepitaxial β-Ga₂O₃ (**b**) films. Red boxes mark isolated emission points. **c** PL spectra fitted to Lorentzian functions for obtaining the individual weightage of zero-phonon line (ZPL) and phonon sideband (PSB) for both homoepitaxial and heteroepitaxial β-Ga₂O₃ films. The peaks marked by square grids and asterisks originate from the Raman mode of β-Ga₂O₃ and sapphire substrate, respectively. Second-order correlation functions $g^2(\tau)$ of the single photon emitters (SPEs) from the homoepitaxial β-Ga₂O₃ (**d**) and the heteroepitaxial β-Ga₂O₃ (**e**) films measured under 5 mW continuous wave laser excitation. **f** Saturation behavior of the emission intensity of SPEs along with a theoretical fit for the homoepitaxial/heteroepitaxial β-Ga₂O₃ films.

follows: β-Ga₂O₃ (−201)[010]//sapphire (0001)[1−100]. The orientation relationship of β-Ga₂O₃ film is also confirmed by the corresponding X-ray diffraction (XRD) patterns (Supplementary Fig. 6a). The XRD rocking curve of (−201) peak displays a full width at half maximum (FWHM) of 0.007° (25 arcsec) and 1.21° for the homoepitaxial and heteroepitaxial β-Ga₂O₃ layers (Supplementary Fig. 6b), respectively, suggesting that the crystalline quality of homoepitaxial β-Ga₂O₃ layer is much higher than that of the heteroepitaxial counterpart.

## Photophysical properties of SPEs in β-Ga₂O₃

The spectroscopy measurements on all β-Ga₂O₃ samples were conducted using a homemade confocal microscopy system equipped with an objective lens with high numerical aperture (NA = 0.9) and a 532 nm continuous-wave laser at room temperature. Several plasma-treatment combined with annealing processes have been attempted to generate isolated point-defects in β-Ga₂O₃. Similar to the as-grown β-Ga₂O₃ films without any treatment, both plasma treatment alone and plasma treatment with air-annealing don't cause PL emission, and only two Raman peaks of β-Ga₂O₃ are observed (Supplementary Fig. 7a). By applying plasma treatment followed by vacuum annealing, the β-Ga₂O₃ film exhibits PL defect-emission with strong fluorescence background (Supplementary Fig. 7b). When these β-Ga₂O₃ films were further annealed in air at 850 °C for 30 min, the fluorescence background is substantially reduced (Supplementary Fig. 7c). Thus, we introduced isolated point-defects in β-Ga₂O₃ films through plasma treatment and activated them using vacuum annealing, and followed by air annealing to eliminate the fluorescence background. Unless otherwise specified, this treatment procedure was employed in all subsequent experiments. The normalized confocal photoluminescence (PL) intensity maps for the homoepitaxial and heteroepitaxial β-Ga₂O₃ films are shown in Fig. 2a,b, respectively, where the local bright spots marked by red boxes are isolated emission points. As revealed by an analysis using Ripley's K function (Supplementary Fig. 8), these emission points are uniformly and randomly distributed in both samples. The density of

SPEs is estimated to be 0.02 counts μm⁻², which is comparable with previously reported quantum emitters in SiN[16], GaN[19] and AlN[22]. The PL spectra of two representative SPEs in the homoepitaxial and heteroepitaxial films are shown in Fig. 2c. Both PL spectra exhibit an asymmetric zero-phonon line (ZPL), possibly due to interactions with phonons[23]. As revealed by the fitting results, the spectra are composed of a prominent ZPL centered at ~565 nm (~2.19 eV) and a lower-intensity phonon sideband (PSB) red-shifted by ~34 meV, as well as a weak peak at ~610 nm due to multi-phonon scattering and/or anharmonic process[35]. The sharp peaks at ~530/550 nm (marked by hash symbols) and 690 nm (marked by asterisk) in the PL spectra originate from the Raman mode of β-Ga₂O₃ and sapphire substrate, respectively. The Debye–Waller (DW) factors (i.e., the probability of coherently emitting into the ZPL) are calculated to be 77% (homoepitaxial) and 70% (heteroepitaxial), which are higher than the values of several commonly SPEs, such as G-center in Si (~15%@10 K)[36], AlN (~29%@10 K)[37], and negatively charged nitrogen vacancy (NV⁻) center in diamond (2.55% @room temperature)[38]. The emitters with a higher DW factor could be directly enhanced via coupling to an optical cavity, achieving better single-photon emission characteristics.

Next, we investigated the single photon purity of the emitters by second-order autocorrelation $g^2(\tau)$ measurements with a Hanbury Brown-Twiss (HBT) interferometer. The normalized second-order correlation function $g^2(\tau)$ of SPEs in the homoepitaxial and heteroepitaxial β-Ga₂O₃ films is shown in Fig. 2d,e, respectively. The experimental $g^2(\tau)$ data were fitted (red lines) using a three-level model without background correction[25,39]: $g^2(\tau) = 1-(1+a)e^{-|\tau|/\tau_1}+be^{-|\tau|/\tau_2}$, where $a$ and $b$ are fitting parameters, $\tau_1$ and $\tau_2$ are the lifetimes of the excited and metastable states, respectively. From the fitting, we obtained the $g^2(\tau)$ value of 0.24 ± 0.08 and 0.32 ± 0.08 at zero delay time for the homoepitaxial and heteroepitaxial β-Ga₂O₃, respectively, which is well below the threshold of 0.5, unambiguously proving its nature of single-photon emission. The real quantum emission purity could be higher than the $g^2(0)$ values shown in Fig. 2d, e since they were obtained

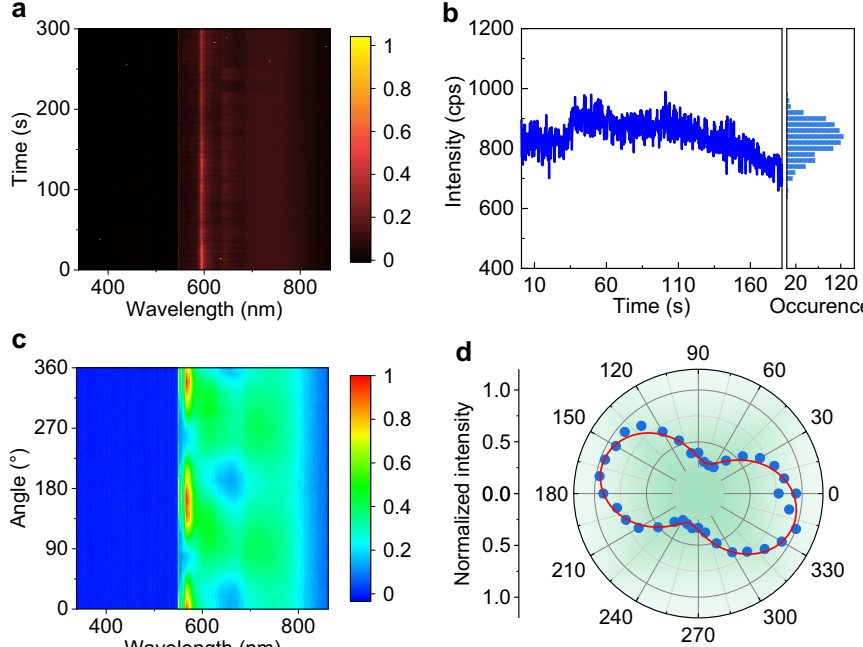

**Fig. 3 | Photostability and polarization of SPEs in homoepitaxial β-Ga₂O₃ films.** **a** Stability tests of the PL spectra of quantum emitters measured under an excitation power of 5 mW and an integration time of 1 s. **b** Time-dependent PL intensity of the emitter with a sampling time bin size of 200 ms, showing no obvious blinking or bleaching. **c** Normalized polarization-dependent contour map of PL spectra for the quantum emitter. **d** Polar coordinate plot of the PL intensity and the polarization angle $\theta$. The data are fitted with a $\cos^2(\theta)$ form fit function, yielding the polarization visibility of 54%.

without any background correction. In addition, because the $g^2(\tau)$ measurements were performed at a high excitation power of ~5 mW, bunching behavior was also observed with a $\tau_2$ of a couple of tens ns.

The emission intensity of SPEs from both β-Ga₂O₃ films as a function of excitation power is presented in Fig. 2f. The saturation data were fitted with the power model: $I(P) = I_\infty \times P/(P + P_{sat})$, where $I_\infty$ and $P_{sat}$ are fitting parameters corresponding to the maximum emission counts and saturation power, respectively[16,19]. The fitted saturation curve yields a saturation count rate of $I_\infty = 1.6 \times 10^4$ counts s⁻¹ and a saturation power of $P_{sat} = 3.3$ mW for the homoepitaxial β-Ga₂O₃. Similarly, we obtain a $I_\infty$ of $6.9 \times 10^3$ counts s⁻¹ and a $P_{sat}$ of 2.4 mW for the heteroepitaxial β-Ga₂O₃ film. These results show that both the emission rate and the purity of single photons emitted from the homoepitaxial β-Ga₂O₃ are higher compared to the heteroepitaxial β-Ga₂O₃, indicating the superiority of high crystallinity of β-Ga₂O₃ as a host of SPEs. It should be noted that the PL spectra were recorded using the charge-coupled device detector of the spectrometer in this work, which leads to a decrease in fluorescence intensity by approximately 50-fold. After correction for the reduced PL intensity, the saturation brightness of β-Ga₂O₃-based SPEs can reach around 10⁵ counts s⁻¹, which is comparable to the other solid-state SPEs[40,41].

Photostability of the quantum emitters was investigated by recording PL spectra under continuous wave excitation and an integration time of 1 s. As shown in Fig. 3a, there is almost no change in the PL intensity and position for more than 300 s of continuous acquisition, demonstrating the long-term stability of the SPEs in the homoepitaxial β-Ga₂O₃. To further examine the blinking and photobleaching, the time-dependent PL intensity of the emitter with a sampling time bin size of 200 ms is displayed in Fig. 3b. The emitter exhibits stable emission without obvious blinking or bleaching over a measurement period of 180 s even under near-saturation excitation power (5 mW), revealing the emitter's absolute photostability at room temperature. The photostability of the quantum emitters hosted by

heteroepitaxial β-Ga₂O₃ is similar to that of its homoepitaxial counterpart with occasional blinking (Supplementary Fig. 9).

We measured the excitation polarization of emitters by rotating a half-wave plate after the polarizer in the excitation path of the confocal set-up while fixing the emission polarization measurements. Figure 3c illustrates the normalized polarization-dependent contour map of PL spectra for the emitter in the homoepitaxial β-Ga₂O₃, and the corresponding waterfall plot of PL spectra is shown in Supplementary Fig. 10. These PL spectra exhibit an obvious periodic variation with the change of polarization angle. Figure 3d shows the corresponding polar coordinate plot of the PL intensity and the polarization angle $\theta$, where the experimental data is perfectly fitted by the function $I(\theta) = I_{min} + I_{max}\cos^2(\theta + \theta_0)$, exhibiting a two-lobed shape. The excitation polarization visibilities $H = (I_{max} - I_{min})/(I_{max} + I_{min})$ is calculated to be 54%, indicating that the quantum emitters in the homoepitaxial β-Ga₂O₃ have a single linearly polarized dipole transition. Similarly, the linear polarization feature is also observed in the heteroepitaxial β-Ga₂O₃ film (Supplementary Fig. 11).

The above results indicate that the quantum emitters can be generated in both homoepitaxial and heteroepitaxial β-Ga₂O₃ films through the plasma treatment and annealing. To examine whether SPEs are prevalent in various types of β-Ga₂O₃, single-crystal β-Ga₂O₃ wafers were also plasma treated and subsequently annealed for further characterization. As shown in Supplementary Fig. 12a, the bright spots marked by red boxes in the PL map are attributed to single-defect centers hosted by the single-crystal β-Ga₂O₃ wafer. The representative PL spectrum consists of a sharp ZPL and a weak and broad PSB (Supplementary Fig. 12b), and a clear antibunching is demonstrated with a very low multiphoton probability of $g^2(0) = 0.18$ (Supplementary Fig. 12c). The SPEs in the single-crystal β-Ga₂O₃ also show high stability (Supplementary Fig. 12d) and linearly polarized characteristics (Supplementary Fig. 12e). The fitted saturation curve yields a saturation emission rate of $I_\infty = 5.7 \times 10^3$ counts s⁻¹ at a saturation power of 4.5 mW (Supplementary Fig. 12f). Apparently, the

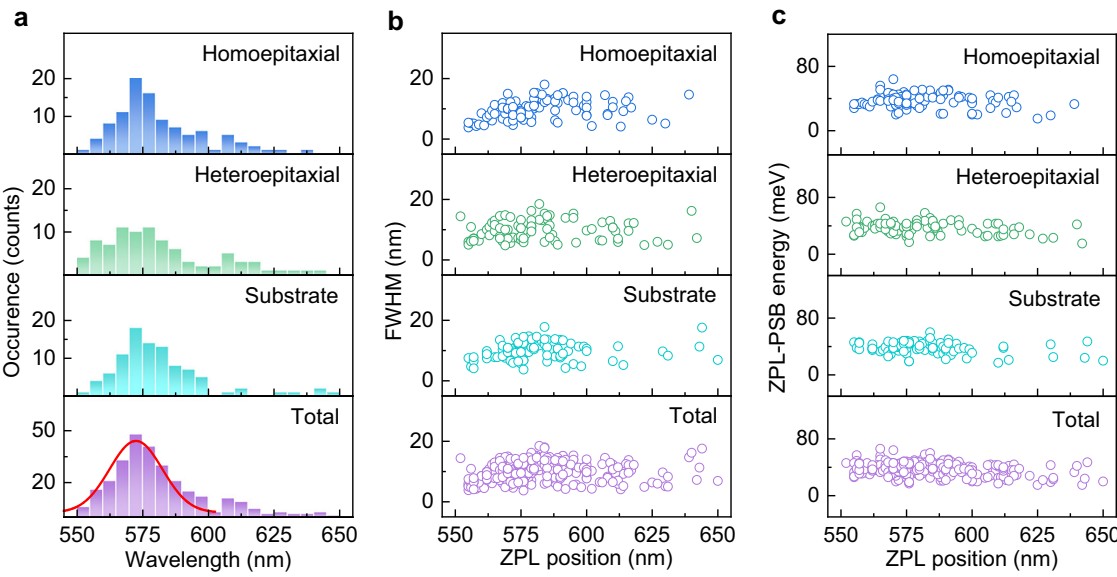

**Fig. 4 | Statistical analysis of quantum emitters in a variety of β-Ga₂O₃. a** Histogram of the ZPL wavelength distribution with a bin width of 5 nm. The red curve represents the Gaussian fitting for ZPL wavelengths. Distribution of the full width at half maximum (FWHM) of ZPL (**b**) and the ZPL-PSB energy separation (**c**).

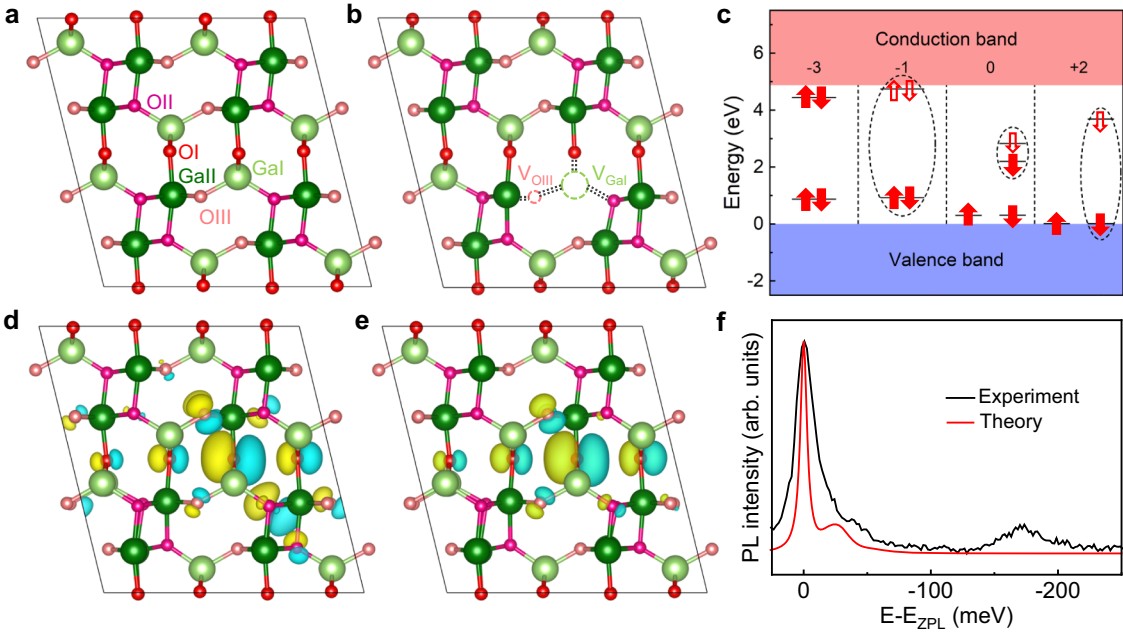

**Fig. 5 | First principal calculations on defects for SPEs. a** The β-Ga₂O₃ supercell on [010] crystal orientation. It is outlined in black lines and contains a total of 160 atoms. Crystallographically distinct Ga and O sites are highlighted in different colors, with $Ga_I$ in light green, $Ga_{II}$ in dark green, $O_I$ in red, $O_{II}$ in magenta, and $O_{III}$ in pink. **b** Schematic structure diagram of the $V_{GaI}$-$V_{OIII}$ defect. **c** Kohn–Sham energy levels of $V_{GaI}$-$V_{OIII}$ defect. The occupied and unoccupied states in the bandgap are depicted as solid and hollow arrows, respectively. To improve clarity, some energy levels are omitted and the complete information is plotted in Supplementary Fig. 15

for reference. Arrow directions signify electron spins, while possible optical transitions are marked by dashed black circles. The wavefunction of the highest occupied states (**d**) and the lowest unoccupied states (**e**) related to optical transitions in neutral $V_{GaI}$-$V_{OIII}$ defects. **f** The calculated PL spectrum (red) for the neutral $V_{GaI}$-$V_{OIII}$ defect is compared with the experimental data (black). To facilitate comparison of the shape and distribution of phonon sidebands, the energy of both spectra is horizontally shifted from ZPL energy.

emitters in single-crystal β-Ga₂O₃ wafers and epitaxial films show similar luminescence characteristics.

To further investigate the statistics of the optical properties of these SPEs, we collected a large number of PL spectra in three series: 101/88 emitters in a homoepitaxial/heteroepitaxial β-Ga₂O₃ film and 95 emitters in a single-crystal β-Ga₂O₃ wafer, and extracted the ZPL positions and linewidths by fitting the spectra with two Lorentzian functions. The histograms in Fig. 4a show the ZPL wavelength

distribution for the emitters in different types of β-Ga₂O₃. These histograms show that the SPE wavelengths vary over a wide range of 550–650 nm, with most of the emitters exhibiting a ZPL around 560–590 nm. Nevertheless, there are some differences between different types of β-Ga₂O₃. For the single-crystal β-Ga₂O₃ wafer and homoepitaxial β-Ga₂O₃ film, about 72% of the emitters have ZPL wavelengths within the range of 575 ± 15 nm, indicating a relatively narrow spectral distribution. By contrast, only 60% of the investigated

**Table 1 | Comparison of the calculated and experimental optical properties for the defect emission**

| type | charge | μ | ZPL@HSE (eV) | $E_{LO}$ (meV) | ΔQ (Å amu$^{1/2}$) | S | DW factor | $\tau_r$ (ns) |
|---|---|---|---|---|---|---|---|---|
| $V_{GaI}$-$V_{OIII}$ | 0 | 1 | 2.34 | 25.4 | 0.353 | 0.28 | 0.76 | 6.42 |
| Exp. | - | - | 2.20 | 27.0 | - | 0.30 | 0.77 | 2.52 |

These properties include the charge state, magnetic moment (μ), zero-phonon line (ZPL), the LO phonon energy ($E_{LO}$), the mass-weighted displacement (ΔQ), electron-phonon coupling effects including the Huang-Rhys (S) factor and Debye-Waller (DW) factor, and radiative lifetime ($\tau_r$).

emitters are located in the spectral range from 560 to 590 nm for the heteroepitaxial β-Ga$_2$O$_3$. Figure 4b summarizes the distribution of the ZPL linewidth of the quantum emitters in three types of β-Ga$_2$O$_3$. The majority of SPEs have a ZPL FWHM less than 15 nm with an average FWHM of 9.7 nm, suggesting the excellent monochromaticity of emitters hosted by β-Ga$_2$O$_3$. Figure 4c shows the distribution of ZPL-PSB energies of the quantum emitters, where the ZPL-PSB energies of the majority SPEs are distributed in the range of 20–60 meV with a mean value of 37 meV. This energy separation is in good agreement with the energies of phonon modes of β-Ga$_2$O$_3$ (20–90 meV) reported in the literatures[42] and our calculations on the phonon density of states (Supplementary Fig. 13). The statistics of Huang–Rhys (S) factor (the number of phonons emitted during vibrational relaxation) are displayed in Supplementary Fig. 14. These S factors are all less than 0.90 with an average value of ~0.50, signifying that pure electronic transitions dominate in β-Ga$_2$O$_3$. The similarity of luminescence characteristic of the SPEs in different β-Ga$_2$O$_3$ samples demonstrates that these defects have similar crystallographic structure.

## Origin of SPEs in β-Ga$_2$O$_3$

To gain insight into the origin of the quantum emitters observed in β-Ga$_2$O$_3$, we conducted systematic theoretical studies of defect structures via density functional theory (DFT). As is well-known, there are two crystallographically inequivalent gallium sites (labelled Ga$_I$ and Ga$_{II}$) and three inequivalent oxygen sites (labelled O$_I$, O$_{II}$, and O$_{III}$) in monoclinic β-Ga$_2$O$_3$ (Fig. 5a). Due to its low symmetry, β-Ga$_2$O$_3$ harbors a diverse array of defects. The pervasiveness of the SPEs in different β-Ga$_2$O$_3$ samples suggests that intrinsic defects rather than extrinsic defects are responsible for the observed quantum emissions. Ga or O interstitials are metastable and are expected to diffuse at higher temperatures. Actually, there is limited experimental evidence for Ga or O interstitials under normal conditions[43]. Therefore, in this work, four main intrinsic vacancy defects, including gallium vacancies ($V_{Ga}$), oxygen vacancies ($V_O$), Ga-O divacancies ($V_{Ga}$-$V_O$), and O-O divacancies ($V_O$-$V_O$) in their neutral, negatively, and positively charged states are considered. The formation energies for various vacancy defects with different charge states as a function of the Fermi energy were calculated previously[44]. Firstly, we calculated the Kohn-Sham (KS) energy levels, ZPL energies, and S factors only for these stable charge states of various defects[44] using the Perdew–Burke–Ernzerhof (PBE) functional[45], as listed in Supplementary Table 1. To rapidly identify potential candidate defects for SPEs, two preliminary screening rules are applied to exclude some defects. (i) Defects capable of emitting single photon usually require the formation of a two-level system between the conduction band minimum (CBM) and the valence band maximum (VBM). These defects without a two-level system are excluded as candidates for SPEs, and they are marked by "×" in Supplementary Table 1. (ii) These defects with calculated S > 1.80 (marked by "-" in Supplementary Table 1) are also dismissed to refine these remaining candidates, since experimentally statistically S factors are all less than 0.90 with an average value of 0.50. Further, we employed the Heyd-Scuseria-Ernzerhof (HSE) hybrid functional[46] to obtain accurate ZPL energies, which is a key predictor to evaluate the origin of the experimentally observed quantum emitters. As summarized in Supplementary Table 1, the predicted ZPL energy (2.34 eV) of neutral $V_{GaI}$-$V_{OIII}$ defects (Fig. 5b) align closely with the experimental values (2.20 eV), indicating a possible SPE candidate. The KS energy levels of

the $V_{GaI}$-$V_{OIII}$ defect in different charge states are shown in Fig. 5c and Supplementary Fig. 15. Among them, when the defect charge state is −3, there is no no-level system suitable for optical transitions. The +2 and −1 charge states are also discarded due to the calculated high S factors. To further identify the candidate defects, the calculated optical properties of the neutral $V_{GaI}$-$V_{OIII}$ defect, including ZPL energy, longitudinal optical (LO) phonon energy ($E_{LO}$), DW and S factors, excited-state lifetime (τ) are compared with the typical experimental results. As seen from Table 1, the theoretically predicted optical properties are fully consistent with the experimental data, indicating that the neutral $V_{GaI}$-$V_{OIII}$ defect is the most promising SPE candidate. The calculated electron-phonon spectral density, from which the S factor and $E_{LO}$ can be derived, is shown in Supplementary Fig. 16.

Furthermore, the predicted excited-state lifetime of the neutral $V_{GaI}$-$V_{OIII}$ is about 1.12 ns, and such short lifetime can be understood by its wavefunctions of the highest occupied orbital and the lowest unoccupied orbital related to optical transitions. As shown in Fig. 5d, e, both wavefunctions show typical localization characteristics, corresponding to deep-level defects. Their similar spatial distribution implies a large transition dipole moment between the two levels, which explains the short excited-state lifetime. Additionally, the calculated emission spectrum of the neutral $V_{GaI}$-$V_{OIII}$ is shown in Fig. 5f, which has been horizontally shifted from the ZPL energy to match the experimental result. The similarity between the theoretical and experimental line shapes is remarkable, especially, the PSB accounts for the asymmetric broadening of the PL spectrum. Based on these calculations, we identified the neutral $V_{GaI}$-$V_{OIII}$ as a suitable defect to explain the observed single-photon emissions in β-Ga$_2$O$_3$.

## Discussion

In summary, optically stable single-photon emissions from point defects in β-Ga$_2$O$_3$ in the visible range have been observed at room temperature. We found that the SPEs can be generated in various β-Ga$_2$O$_3$, including homoepitaxial and heteroepitaxial β-Ga$_2$O$_3$ films and commercially available single-crystal β-Ga$_2$O$_3$ wafers, by plasma treatment combined with annealing. Compared with the heteroepitaxial β-Ga$_2$O$_3$ films, the emitters hosted by homoepitaxial β-Ga$_2$O$_3$ films and single-crystal β-Ga$_2$O$_3$ wafers exhibit better performance. Photophysical analysis reveals bright (~10$^5$ counts/s after correction), pure ($g^2(0) < 0.2$), stable, linearly polarized room-temperature quantum light emission from color centers in various β-Ga$_2$O$_3$ samples. DFT calculations indicate that the neutral $V_{GaI}$-$V_{OIII}$ defect is responsible for the observed single-photon emissions. This bright SPEs operating at room temperature provides a fundamental building block for β-Ga$_2$O$_3$-based optoelectronic devices and future integrated quantum photonics.

## Methods
### Sample fabrication
The homoepitaxial and heteroepitaxial β-Ga$_2$O$_3$ thin films were grown on the β-Ga$_2$O$_3$ (−201) single crystal substrates and sapphire (0001) substrates by LPCVD, respectively. The LPCVD system consists of a two-inch tubular quartz reactor and has two independently controlled temperature zones. Before the growth process, single-side polished β-Ga$_2$O$_3$ (−201) substrates (1 mm thick) and sapphire (0001) substrates (0.5 mm thick) were sequentially cleaned by acetone, ethanol, and deionized water in an ultrasonic bath, followed by drying with N$_2$ gas.

High purity Ga pellets and $O_2$ with a flow rate of 10 sccm were used as precursors. The Ga-source and substrate temperatures were fixed at 830 and 800 °C, respectively. Ar with a flowing at 50 sccm acted as the carrier gas, and the growth was conducted at a pressure of 40 Pa for 30 min. Finally, the furnace was cooled down to room temperature in an Ar atmosphere.

The plasma treatment was carried out in a home-made system equipped with a 13.56 MHz radio frequency (RF) plasma generator. The cleaned β-$Ga_2O_3$ samples were first loaded into the plasma treatment chamber, then it was evacuated to a pressure of less than 1 Pa. Afterward, the plasma treatment was performed at room temperature and a constant RF power of 100 W for 10 min. Finally, the plasma-treated β-$Ga_2O_3$ samples were annealed in a tube furnace at 850 °C and an atmospheric pressure of less than 1 Pa for 30 min to promote the formation of optically active defects. A further annealing at 850 °C under air atmosphere for 30 min is usually performed to reduce the influence of spectral background fluorescence.

## Characterization

The surface morphology of β-$Ga_2O_3$ samples was characterized by atomic force microscope (AFM) with a NT-MDT solver P47 microscope in the semi-contact mode. Raman spectra were collected by a Smart-Raman confocal-micro Raman system (Horiba iHR550 spectrometer) under the backscattering geometry with a 532 nm laser in the region of $100-800 \, cm^{-1}$. XRD measurements were carried out with a Rigaku D/MAX-2500 diffractometer with a Cu Kα ($\lambda = 1.5406$ Å) radiation source operating at 40 kV and 40 mA. X-ray photoelectron spectroscopy (XPS) core levels were acquired with an ESCALAB 250Xi spectrometer using monochromatized Al Kα source (1486.6 eV). The microstructures of epitaxial β-$Ga_2O_3$ films were analyzed by transmission electron microscopy (TEM) using a Talos F200S microscope operating at 200 kV.

The optical spectroscopy measurements on all β-$Ga_2O_3$ samples were conducted using a custom-built micro-area confocal PL setup equipped with an objective lens of high numerical aperture (0.9) and a 532 nm continuous-wave (CW) laser at the room temperature. PL spectra were acquired with the charge-coupled device detector of Andor Kymera 328i-B1 spectrometer and a CCD camera, resulting in an approximate 50-fold reduction in fluorescence intensity. The excitation polarizations were measured by placing a broadband polarizer (Thorlabs) and a half-wave plate in the excitation path. Second-order autocorrelations ($g^2(\tau)$) were measured using a Hanbury Brown-Twiss system excited by a 532 nm continuous wave laser and connected to a time-correlated single photon counting module with two avalanche photodiodes (APDs).

## DFT calculations

DFT calculations were performed using the projector augmented wave (PAW) method as implemented in the Vienna Ab initio Simulation Package (VASP)[47]. For defect modeling, a 160-atom supercell was constructed, and self-consistent calculations were carried out with a single Γ-point. The energy convergence criterion and force convergence criterion for structural relaxation were set to 0.02 eV/Å and $10^{-5}$ eV, respectively. All DFT calculations utilized a plane-wave cutoff energy of 520 eV. To predict optical properties, we calculated $\Delta Q$ and $S$ using constrained DFT based on the PBE ground-state and excited-state geometries. The ΔSCF method was employed to predict ZPL energies under both PBE and HSE functionals. By adjusting $\alpha$ to 0.39, we calculated the bandgap of bulk β-$Ga_2O_3$ as 4.87 eV, which is in excellent agreement with experimental values[28]. The radiative lifetime $\tau_r$ associated with ZPL was determined using Fermi's golden rule[48]. PL line shapes were simulated using the PyPhotonics package[49], and phonon energies were determined via the finite displacement method implemented in the Phonopy code[50].

## Data availability

The data that support the findings of this study are available within in the published article and its Supplementary Information. Any other relevant data are available from the corresponding authors upon request.

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

## Acknowledgements

This work was partially supported by the National Natural Science Foundation of China (Grant No. 62174009, J.M. and 52473268, X.Z.) and the National Key Research and Development Program of China (Grant No. 2024YFA1409702, X.Z.).

## Author contributions

X.Z., J.M., Y.S. and Z.X. conceived the idea. X.Z. and J.M. directed and supervised the project. Y.S. carried out the sample fabrication and characterizations. Z.X. performed the DFT calculations. L.Z. helped the optical spectroscopy measurements. J.J., A.W., Z.L., H.Y. and Z.Y. were involved in data analysis. Y.S. drafted the manuscript with assistance from Z.X., J.M. and X.Z. Revised the manuscript. All authors have read the results and commented upon or contributed to discussions and finalizing the manuscript.

## Competing interests

The authors declare no competing interests.
