## [Transparent Peer Review file · Nature Communications]

Room-temperature single-photon emission from β -Ga₂O₃

Corresponding Author: Professor Xingwang Zhang

Version 0:

Reviewer comments:

Reviewer #1

(Remarks to the Author)

The study reports the discovery of room-temperature single-photon emitters in the emerging semiconductor Ga₂O₃. The authors observe visible single-photon emission in various samples following plasma and annealing treatments, and employ theoretical modeling to identify the intrinsic defect responsible for this behavior. These findings are of significant importance to the field and are likely to stimulate a new line of research on Ga₂O₃ for quantum applications.

The nature of the work raises questions regarding the robustness of the single-photon emitters and the role of the sample treatments in enabling them. It is essential that the authors present the luminescence spectra of the samples prior to any treatment. Additionally, it would be valuable to discuss the parameter space for engineering these single-photon emitters—for instance, how the emission properties depend on annealing conditions.

Here are a few minor points:

- Given that the films are only 200 nm thick and that photoluminescence is done below bandgap excitation, how confident are the authors that the luminescence comes from the film and not the substrate?
- RMS roughness values in page 3 should be given with error bars.
- I would recommend avoiding words like “obviously”
- “Crystallo-graphic” should be written “crystallographic”
- Figure 3c, the y-axis should be “Angle”
- Figure 4c would benefit to have the y-axis in meV
- Page 10, line 275: the sentence “Actually, ... conditions.” doesn’t make sense grammatically

Reviewer #2

(Remarks to the Author)

In the manuscript titled “Room-temperature single-photon emission from β -Ga₂O₃”, Shi et al demonstrate the first reports of single photon emitters in β -Ga₂O₃ in homoepitaxial, heteroepitaxial and bulk samples. Photophysical measurements similar to other such works in literature are presented to an acceptable standard to definitively identify this new SPE. Density functional theory evidence is presented to offer a physical origin for the SPE with a convincing conclusion. The manuscript is well written and features sufficient references and I believe this result will be of interest to those in the wider research community.

Subject to the following amendments being satisfied, it is my recommendation that this work is published.

On line 130, the authors make reference to the spatial distribution of emitters being uniform and random. Whilst not heavily examined in similar works, this point can be reinforced by an analysis using Ripley’s K function to determine if the emitters are clustered or diffuse.

The autocorrelation functions presented in Figure 2. D and E, are somewhat noisy. Could the authors experiment with rebinning the data and quote the error in $g^{(2)}(0)$ to ensure that these results are consistent with single-photon emission?

Version 1:

Reviewer comments:

Reviewer #1

(Remarks to the Author)

Thank you for addressing my comments adequately.

Reviewer #2

(Remarks to the Author)

I thank the authors for taking my comments into consideration. I am satisfied my comments have been sufficiently addressed and can now fully recommend this for publication in Nature Comms.

Point-by-Point Response to the Reviewers' Comments for NCOMMS-25-61281

Dear Editors and Reviewers:

We would like to thank the Reviewers for their insightful comments and valuable suggestions on our manuscript entitled "Room-temperature single-photon emission from β -Ga₂O₃". Those comments are all valuable and very helpful for revising and improving our paper, as well as the important guiding significance to our researches. We have made careful revision of our manuscript based on comments suggested by the Reviewers. Below is our point-by-point response to these comments. For ease of tracking, the reviewers' comments are reproduced by verbatim in italics, followed with our responses in purple and the revisions in blue.

Reviewer #1 Comments:

The study reports the discovery of room-temperature single-photon emitters in the emerging semiconductor Ga₂O₃. The authors observe visible single-photon emission in various samples following plasma and annealing treatments, and employ theoretical modeling to identify the intrinsic defect responsible for this behavior. These findings are of significant importance to the field and are likely to stimulate a new line of research on Ga₂O₃ for quantum applications. The nature of the work raises questions regarding the robustness of the single-photon emitters and the role of the sample treatments in enabling them.

Response: We are very grateful for your thoughtful and positive assessment of our manuscript. We sincerely thank the reviewer for the recognition of our demonstration of room-temperature photostable single-photon emission from β -Ga₂O₃. According to your valuable suggestions, we have made a few corrections to our previous draft as listed below.

Comment #1-1: *It is essential that the authors present the luminescence spectra of the samples prior to any treatment.*

Response on comment #1-1: Many thanks for the reviewer's valuable comments. We agree that it is necessary to display the PL spectra of β -Ga₂O₃ thin films without treatment. According to the reviewer's suggestion, the PL spectrum of the homoepitaxial β -Ga₂O₃ films without any treatment has been added in the revised manuscript as Supplementary Fig. 7a, in which only two Raman peaks from β -Ga₂O₃ were observed. It can be concluded

that the single photon emitter is introduced by plasma treatment and subsequent annealing process.

Supplementary Fig. 7a | The PL spectra of homoepitaxial β -Ga₂O₃ thin films under different treatment conditions (As-grown sample, plasma treatment alone, plasma treatment + air-annealing). The peaks marked by # originate from the Raman mode of the β -Ga₂O₃.

Our revision to the manuscript:

Page 5, lines 128-131: “Similar to the as-grown β -Ga₂O₃ films without any treatment, both plasma treatment alone and plasma treatment with air-annealing don’t cause PL emission, and only two Raman peaks of β -Ga₂O₃ are observed (Supplementary Fig. 7a).” has been added.

Comment #1-2: *Additionally, it would be valuable to discuss the parameter space for engineering these single-photon emitters—for instance, how the emission properties depend on annealing conditions.*

Response on comment #1-2: Thanks for the reviewer’s valuable suggestions. According to the reviewer’s suggestion, the parameter spaces for engineering these single-photon emitters have been discussed in the revised manuscript. As shown in Supplementary Fig. 7a, the individual plasma treatment does not cause any PL defect-emission in β -Ga₂O₃, and the situation remains similar even with subsequent air annealing. To activate the defects induced by plasma treatment, the plasma treated β -Ga₂O₃ was annealed at 850 °C for 30 min under a pressure of <1 Pa (vacuum annealing), leading to PL defect-emissions with strong background fluorescence (Supplementary Fig. 7b). In order to suppress the background fluorescence, the β -Ga₂O₃ film after plasma treatment and vacuum annealing was further annealed in air at 850 °C for 30 min. Indeed, the PL defect-emissions with a

substantially reduced fluorescence background is clearly observed for the β -Ga₂O₃ after plasma treatment combined with vacuum annealing and air annealing (Supplementary Fig. 7c). Furthermore, the plasma treatment power of 100-150 W is appropriate as indicated by the stronger PL emission.

Our revision to the manuscript:

Page 5, lines 127-138: “Several plasma-treatment combined with annealing processes have been attempted to generate isolated point-defects in β -Ga₂O₃. Similar to the as-grown β -Ga₂O₃ films without any treatment, both plasma treatment alone and plasma treatment with air-annealing don’t cause PL emission, and only two Raman peaks of β -Ga₂O₃ are observed (Supplementary Fig. 7a). By applying plasma treatment followed by vacuum annealing, the β -Ga₂O₃ film exhibits PL defect-emission with strong fluorescence background (Supplementary Fig. 7b). When these β -Ga₂O₃ films were further annealed in air at 850 °C for 30 min, the fluorescence background is substantially reduced (Supplementary Fig. 7c). Thus, we introduced isolated point-defects in β -Ga₂O₃ films through plasma treatment and activated them using vacuum annealing, and followed by air annealing to eliminate the fluorescence background. Unless otherwise specified, this treatment procedure was employed in all subsequent experiments.” has been added.

Supplementary Fig. 7 | The PL spectra of homoepitaxial β -Ga₂O₃ thin films under different treatment conditions. a As-grown sample, plasma treatment alone, plasma treatment + air-annealing. **b** Plasma treatment + vacuum-annealing. **c** Plasma treatment + vacuum-annealing + air-annealing under different plasma powers. The peaks marked by # originate from the Raman mode of the β -Ga₂O₃.

Comment #1-3: Given that the films are only 200 nm thick and that photoluminescence is done below bandgap excitation, how confident are the authors that the luminescence comes from the film and not the substrate?

Response on comment #1-3: Many thanks for the reviewer’s valuable question. As discussed in Response on comment #1-2, the PL emission of β -Ga₂O₃ is generally believed to come from point-defects caused by plasma treatment. The physical effect depth of plasma treatment is relatively shallow, typically ranging from 1 to several tens of nanometers, mainly due to the ion bombardment on the material surface. Since the thickness of our β -Ga₂O₃ films is about 200 nm, which is much larger than the depth of plasma treatment, we believe that the single photon emitters generated in both homoepitaxial and heteroepitaxial β -Ga₂O₃ samples come from the epitaxial film rather than the substrate.

Comment #1-4: RMS roughness values in page 3 should be given with error bars.

Response on comment #1-4: Many thanks for the reviewer’s suggestions. According to the reviewer’s suggestion, AFM images of the single crystal β -Ga₂O₃ wafer, the homoepitaxial β -Ga₂O₃ and heteroepitaxial β -Ga₂O₃ were measured at five positions to obtain the average value and the standard deviation (SD) of RMS roughness (Supplementary Fig. 1). Then, the RMS roughness values in Page 3 are given by the average value with error bars.

Supplementary Fig. 1 | AFM images of the β -Ga₂O₃. **a1-a5** Single crystal β -Ga₂O₃ wafer. **b1-b5** Homoepitaxial β -Ga₂O₃ films. **c1-c5** Heteroepitaxial β -Ga₂O₃ films.

Our revision to the manuscript:

Page 3, line 79: “root mean square (RMS) roughness of 0.47 nm.” has been changed to “root mean square (RMS) roughness of 0.48 ± 0.04 nm (Supplementary Fig. 1a)”. **line 81:** “RMS roughness of 2.4 nm” has been changed to “RMS roughness of 2.5 ± 0.31 nm (Supplementary Fig. 1b)”. **lines 84-85:** “RMS roughness of 2.8 nm” has been changed to “RMS roughness of 3.1 ± 0.22 nm (Supplementary Fig. 1c)”

Comment #1-5: I would recommend avoiding words like “obviously”

Response on comment #1-5: Many thanks for the reviewer’s suggestions. The sentences have been revised to avoid using the word “obviously”.

Our revision to the manuscript:

Page 5, line 141: “Obviously,” has been changed to “As revealed by an analysis using Ripley’s K function (Supplementary Fig. 8),”.

Page 8, line 228: “Obviously,” has been changed to “indicating that ...”

Comment #1-6: Crystallo-graphic” should be written “crystallographic”

Response on comment #1-6: Many thanks for the reviewer’s suggestion. All words “crystallo-graphic” have been written as “crystallographic” in the revised manuscript.

Comment #1-7: Figure 3c, the y-axis should be “Angle”

Response on comment #1-7: Many thanks for the reviewer’s kind remind. The y-axis of Figure 3c has been revised as “Angle”.

Comment #1-8: Figure 4c would benefit to have the y-axis in meV

Response on comment #1-8: Many thanks for the reviewer’s helpful suggestions, the y-axis in Figure 4c is represented as “meV” in the revised manuscript.

Fig. 4 | Statistical analysis of quantum emitters in a variety of β -Ga₂O₃. c Distribution of the ZPL-PSB energy separation.

Comment #1-9: Page 10, line 275: the sentence “Actually, ... conditions.” doesn’t make sense grammatically

Response on comment #1-9: Many thanks for the reviewer’s comments. The related sentence has been rewritten to correct grammatical errors.

Our revision to the manuscript:

Page 10, lines 286-287: “Actually, there is limited experimental evidence for under normal conditions.” has been changed to “**Actually, there is limited experimental evidence for Ga or O interstitials under normal conditions.**”

Reviewer #2 Comments:

In the manuscript titled “Room-temperature single-photon emission from β -Ga₂O₃”, Shi et al demonstrate the first reports of single photon emitters in β -Ga₂O₃ in homoepitaxial, heteroepitaxial and bulk samples. Photophysical measurements similar to other such works in literature are presented to an acceptable standard to definitively identify this new SPE. Density functional theory evidence is presented to offer a physical origin for the SPE with a convincing conclusion. The manuscript is well written and features sufficient references and I believe this result will be of interest to those in the wider research community. Subject to the following amendments being satisfied, it is my recommendation that this work is published.

Response: We sincerely appreciate the reviewer’s positive assessment of the significance of our work. We are delighted that the reviewer found our progress meaningful, the manuscript well-written, and the references sufficient, as well as the potential applications diverse. We are also grateful for the reviewer’s insightful comments and constructive suggestions, and we have carefully addressed each point in the following responses to further improve the manuscript.

Comment #2-1: *On line 130, the authors make reference to the spatial distribution of emitters being uniform and random. Whilst not heavily examined in similar works, this point can be reinforced by an analysis using Ripley’s K function to determine if the emitters are clustered or diffuse.*

Response on #2-2: Many thanks for the reviewer’s valuable suggestion. According to the reviewer’s suggestion, the distribution of emitters in homoepitaxial and heteroepitaxial β -Ga₂O₃ thin films was analyzed using the Ripley’s K-function. As shown in Supplementary Fig. 8, the observed K values consistently fluctuating within the confidence interval, indicating that the emitters are randomly distributed and do not exhibit obvious clustering or dispersion trends.

Our revision to the manuscript:

Page 5, lines 141-142: “Obviously, these emission points are uniformly and randomly distributed in both samples.” has been changed to “As revealed by an analysis using Ripley’s K function (Supplementary Fig. 8), these emission points are uniformly and randomly distributed in both samples.”

Supplementary Fig. 8 | Distribution of emitters in the homoepitaxial and heteroepitaxial β -Ga₂O₃ thin films. PL intensity maps of the homoepitaxial β -Ga₂O₃ (a) and heteroepitaxial β -Ga₂O₃ (c) thin films. The corresponding Ripley's K function of emitters for the homoepitaxial β -Ga₂O₃ (b) and heteroepitaxial β -Ga₂O₃ (d) thin films. The observed K values consistently fluctuating within the confidence interval, indicating that the emitters are randomly distributed and do not exhibit obvious clustering or dispersion trends.

Comment #2-2: The autocorrelation functions presented in Figure 2. D and E, are somewhat noisy. Could the authors experiment with rebinning the data and quote the error in $g^{(2)}(0)$ to ensure that these results are consistent with single-photon emission?

Response on #2-2: Many thanks for the reviewer's helpful suggestions. Indeed, the autocorrelation function $g^2(\tau)$ shown in Fig. 2d,e is a little noisy. To improve the signal-to-noise ratio of data, the $g^2(\tau)$ were remeasured by increasing measurement time and manipulated by rebinning the data. To ensure that these results are consistent with single-photon emission, the error in $g^2(0)$ has also been quoted. From the fitting, we obtained the $g^2(\tau)$ at zero delay time of $g^2(0)=0.24\pm 0.08$ for the homoepitaxial β -Ga₂O₃ and $g^2(0)=0.32\pm 0.08$ for the heteroepitaxial β -Ga₂O₃, respectively. Both $g^2(0)$ values are well below the threshold of 0.5, unambiguously proving its nature of single-photon emission.

Fig. 2 | Characterization of room-temperature SPEs in β -Ga₂O₃. **d,e** Second-order correlation functions $g^2(\tau)$ of the SPEs from the homoepitaxial β -Ga₂O₃ (**d**) and the heteroepitaxial β -Ga₂O₃ (**e**) films measured under 5 mW continuous wave laser excitation.

Our revision to the manuscript:

Page 6, lines 165-167: “From the fitting, we obtained the $g^2(\tau)$ value of 0.23 and 0.35 at zero delay time for the homoepitaxial and heteroepitaxial β -Ga₂O₃, respectively, ...” has been changed to “From the fitting, we obtained the $g^2(\tau)$ value of 0.24 ± 0.08 and 0.32 ± 0.08 at zero delay time for the homoepitaxial and heteroepitaxial β -Ga₂O₃, respectively, ...”